# Dynamics of Endogenous Auxin and Its Role in Somatic Embryogenesis Induction and Progression in Cork Oak

**DOI:** 10.3390/plants12071542

**Published:** 2023-04-03

**Authors:** Elena Carneros, Jorge Sánchez-Muñoz, Yolanda Pérez-Pérez, Beatriz Pintos, Aránzazu Gómez-Garay, Pilar S. Testillano

**Affiliations:** 1Pollen Biotechnology of Crop Plants Group, Biological Research Center Margarita Salas, CIB-CSIC, Ramiro de Maeztu 9, 28040 Madrid, Spain; ecarneros@cib.csic.es (E.C.); jsanchezmjob@gmail.com (J.S.-M.); yperez@cib.csic.es (Y.P.-P.); 2Department of Genetics, Microbiology and Physiology, Complutense University of Madrid, 28040 Madrid, Spain; bpintos@bio.ucm.es (B.P.); magom02@bio.ucm.es (A.G.-G.)

**Keywords:** anti-IAA, auxin, forest species, immunolocalization, kynurenine, RT-qPCR

## Abstract

Somatic embryogenesis (SE) is a feasible in vitro regeneration system with biotechnological applications in breeding programs, although, in many forest species, SE is highly inefficient, mainly due to their recalcitrance. On the other hand, SE represents a valuable model system for studies on cell reprogramming, totipotency acquisition, and embryogenic development. The molecular mechanisms that govern the transition of plant somatic cells to embryogenic cells are largely unknown. There is increasing evidence that auxins mediate this transition and play a key role in somatic embryo development, although data on woody species are very limited. In this study, we analyzed the dynamics and possible role of endogenous auxin during SE in cork oak (*Quercus suber* L.). The auxin content was low in somatic cells before cell reprogramming, while it increased after induction of embryogenesis, as revealed by immunofluorescence assays. Cellular accumulation of endogenous auxin was also detected at the later stages of somatic embryo development. These changes in auxin levels correlated with the expression patterns of the auxin biosynthesis (*QsTAR2*) and signaling (*QsARF5*) genes, which were upregulated after SE induction. Treatments with the inhibitor of auxin biosynthesis, kynurenine, reduced the proliferation of proembryogenic masses and impaired further embryo development. *QsTAR2* and *QsARF5* were downregulated after kynurenine treatment. Our findings indicate a key role of endogenous auxin biosynthesis and signaling in SE induction and multiplication, as well as somatic embryo development of cork oak.

## 1. Introduction

Cork oak (*Quercus suber* L.) is one of the most characteristic woody species of the Mediterranean ecosystem, with great ecological and economic value. One of its most distinctive characteristics is cork production. Cork is traditionally used in wine bottling, but is also appreciated in other applications, such as thermal and acoustic insulation in construction. Moreover, its acorns are used for cattle feed in agroforestry systems, mainly for farming of the Iberian pig breed. Cork oak is threatened by various causes, such as illnesses, forest fires, and difficult natural regeneration, justifying the improvement and conservation programs of this species. Classical genetic breeding programs have important limitations in forest trees due to their long reproductive cycles and the difficulty of seed conservation and vegetative reproduction [1,2]. SE is considered as the most suitable in vitro regeneration system, and has become a useful biotechnological tool for plant breeding, propagation, and conservation strategies [3,4,5]. SE is a powerful technique, as it allows for the practical application of different biotechnological techniques such as large-scale propagation of selected material, cryopreservation of elite genotypes, transformation, and gene editing [6,7,8,9]. In cork oak, SE has been achieved in several in vitro embryogenesis systems using different explants, such as immature zygotic embryos, anthers, or leaves [10,11,12]. Despite the great potential of SE in forest species, many of them present low and variable efficiency, thus limiting the use of this technique, since the mechanisms that control the cellular processes underlying SE remain elusive [13,14,15]. An understanding of the regulatory network of cell reprogramming and embryogenesis initiation would allow for its efficient manipulation, and, therefore, the optimization of protocols. Several reports have proposed that hormones are crucial factors underlying totipotency acquisition and embryogenesis initiation [16,17,18] in herbaceous species [19], as well as in woody species [20,21].

Auxin is the most significant phytohormone involved in plant development [22,23]. Moreover, auxin plays a regulatory role in cell division, expansion, and differentiation [24,25,26]. Indole-3-acetic acid (IAA), the main active form of auxin, has been described to play a crucial role in zygotic embryo development from the early stages [27] to embryo polarization and differentiation, and an upregulation of auxin biosynthesis has been detected throughout embryo formation [28,29]. The main pathway of auxin biosynthesis is the IPA (indole-3-pyruvic acid) route [30,31]. This route comprises two steps: first, the amino acid L-tryptophan is converted in IPA by TRYPTOPHAN AMINO TRANSFERASE OF ARABIDOPSIS 1 (TAA1) and its related proteins TRYPTOPHAN AMINO TRANSFERASE-RELATED 1 and 2 (TAR1, TAR2); then, IPA is converted to IAA by flavin monooxygenases encoded by the *YUCCA* (*YUC*) gene family [32,33]. It is well known that the *TAA1/TAR* and *YUC* genes play an important role in embryogenesis [28,30,31]. Recently, it has been reported that auxin biosynthesis is essential for the maintenance of embryonic cell identity and promotes SE development in Arabidopsis [34]. Auxin can activate both broad and specific transcriptional responses [35]. The central components of the auxin signaling pathway are the TRANSPORT INHIBITOR RESISTANT 1/AUXIN SIGNALING F-BOX (TIR1/AFB) F-box proteins, the AUXIN/INDOLE-3-ACETIC ACID (Aux/IAA) transcriptional co-regulators, and the AUXIN RESPONSE FACTOR (ARF) transcription factors. Auxin promotes an interaction between TIR1/AFB and Aux/IAA, resulting in degradation of the Aux/IAA. The removal of Aux/IAA releases ARF repression. Then, ARFs specifically bind to the promoters of auxin-responsive genes to activate or inhibit the expression of downstream genes [36]. Gene expression associated with ARF activation has been described to be implicated in zygotic embryo development [37] and in SE induction in Arabidopsis [38].

There is increasing evidence that auxins mediate the transition of somatic cells into embryogenic cells [39]. Endogenous auxin content, as well as the application of exogenous auxins, are determining factors for the induction of SE [40]. Several studies have described the involvement of endogenous auxin in microspore reprogramming and in vitro embryo formation in the herbaceous crop species *Brassica napus* and *Hordeum vulgare* [41,42,43]. In tree species, endogenous IAA levels have been shown to be higher during the proliferation of embryogenic masses, as well as in somatic embryos of *Picea abies*, *Abies alba,* and *Quercus alba* [44,45,46]. In *Quercus suber*, endogenous IAA quantified by HPLC showed high levels in fully developed somatic embryos, but decreased during embryo maturation and germination [47]. In this species, IAA has also been localized in proembryos derived from microspores or immature zygotic embryos [48]. However, there is scarce information about the endogenous auxin accumulation, its dynamics and role during SE induction, and further stages of SE development in cork oak. For a better understanding of the role of auxin in plant development, the use of small molecules with inhibitory effects are of great importance. Inhibition of auxin transport and activity, as in N-1-naphthylphthalamic acid (NPA) and α-(*p*-chlorophenoxy)-isobutyric acid (PCIB), has been used classically to analyze the possible role of auxin in several developmental processes, including SE [42,48,49]. Besides these small molecules, kynurenine has been described as an inhibitor of auxin biosynthesis. Kynurenine competitively inhibits TAA1/TAR activity, resulting in the inhibition of IAA biosynthesis [50] and, in consequence, provoking alterations in the developmental processes where auxin is involved [51,52]. However, little is known about the effect of kynurenine on SE.

In the present work, we analyzed auxin content before SE induction and its dynamics over the course of SE in cork oak by means of immunolocalization assays with the monoclonal antibody to IAA. In order to evaluate the involvement of auxin in the process, the effect of the inhibitor of auxin biosynthesis kynurenine on SE cultures was also evaluated. Furthermore, the expression patterns of genes encoding auxin biosynthesis and signaling pathways, *QsTAR2* and *QsARF5*, were analyzed before and during SE, as well as after kynurenine treatments. Our findings indicate the involvement of endogenous auxin biosynthesis and signaling in SE induction and multiplication, as well as in the embryo development of cork oak.

## 2. Results

### 2.1. Auxin Localization and Accumulation during SE

SE was induced from immature zygotic embryos (Figure 1A) randomly collected from several trees in the field. Immature zygotic embryos were cultured in a medium containing 2,4-D for 1 month, and then they were transferred to an auxin-free medium for SE induction. After induction, embryos were produced, either directly from the initial explant or indirectly from proembryogenic masses (PEMs) that were previously formed from explants (Figure 1B). SE cultures of cork oak presented asynchronous development, as different structures corresponding to various developmental stages could be found at the same time point in culture plates (Figure 1C). PEMs appeared in clusters of rounded masses of cellular aggregates, which mostly consisted of proliferating embryogenic cells (Figure 1D). Embryogenic cells of PEMs then either continued proliferating to form new PEMs or proceeded to form somatic embryos (Figure 1C). Under in vitro culture conditions, somatic embryos were continuously developing, producing globular, heart (Figure 1E), torpedo (Figure 1F), and cotyledonary embryos (Figure 1G) which could be observed together with new PEMs in the culture plates (Figure 1C). Recurrent embryogenesis was also observed; PEMs and embryos at different developmental stages produced new PEMs and embryos. Spontaneously, some cotyledonary embryos started to accumulate reserve nutrients in cotyledons, thus increasing in size, becoming opaque and ivory-colored, and giving rise to mature somatic embryos (Figure 1G).

Microscopic analysis revealed that in the initial explant, before cell reprogramming, the immature zygotic embryo (Figure 2A) showed differentiated cells with very large vacuoles that occupied most of the cellular volume, as well as small nuclei located at the cell periphery (Figure 2B). After SE induction, PEMs were formed by aggregates of small embryogenic cells which appeared in clusters, forming proembryos and globular embryos (black arrows in Figure 2C). These young embryos were able to develop into heart somatic embryos (white arrow in Figure 2C). The embryogenic cells showed typical structures, with large central nuclei and prominent nucleoli, low vacuolation, and a high nucleus/cytoplasm volume ratio (Figure 2D). Somatic embryos at more advanced developmental stages, such as early cotyledonary embryos (Figure 2E), presented two different kind of cells: meristematic cells, located at the shoot apical meristem (SAM) and the root apical meristem (RAM) (Figure 2F), and cortex-differentiated cells (square in Figure 2E). Meristematic cells presented the characteristic structure of proliferating cells (Figure 2G). In contrast, cortex cells were much larger, with large vacuoles that occupied most of the cell volume, and small nuclei located at the cell periphery (Figure 2H, open arrow). At the periphery of early cotyledonary embryos, the differentiating epidermis was observed in transverse sections as a single cell layer of small polygonal cells (Figure 2H, black arrow).

To analyze the distribution patterns of endogenous auxin and its cellular accumulation during cork oak SE, immunofluorescence assays were performed using a monoclonal antibody to IAA. The experiments were analyzed by confocal microscopy, keeping the same excitation and emission capture settings for all samples. Before induction, the immature zygotic embryo cells (Figure 3A) showed low or no auxin signaling (Figure 3A’). On the contrary, after induction, PEMs (Figure 3B) showed intense immunofluorescence signaling in the clusters of embryogenic cells inside the masses (Figure 3B’), indicating auxin accumulation in the cytoplasm and nuclei of proliferating embryogenic cells. However, highly vacuolated cells of the PMEs that surrounded the embryogenic cell clusters did not show significant IAA signaling (Figure 3B’, arrow).

In more advanced stages of development, as evidenced by heart and torpedo embryos, auxin accumulation increased, showing greater immunofluorescence intensity (Figure 4A,A’). The meristematic regions of advanced embryos displayed intense IAA labeling (Figure 4B,B’). Conversely, very low or no labeling was detected in highly vacuolated-differentiated cells of the cortex of these developed embryos (Figure 4C,C’). The controls avoiding the primary antibody did not show labeling in any of the cell compartments at any developmental stage of SE (Figure 4A”), supporting the specificity of the immunofluorescence results and indicating that the samples did not exhibit autofluorescence in any subcellular structure.

### 2.2. Effect of the Inhibitor of Auxin Biosynthesis Kynurenine on SE

To assess the possible role of auxin in the progression of cork oak SE, treatments were performed with the small molecule kynurenine. PEMs were selected from in vitro proliferative embryogenic cultures and transferred either to a control medium or a medium containing 200 or 400 µM kynurenine (Figure 5A). The concentrations were selected based on previous studies using kynurenine on microspore embryogenesis in other SE cell cultures [43], although in cork oak, the concentrations were around 10 times higher since this system used gelled media, which present lower diffusion and limited component availability compared to microspore liquid media [43]. The proliferation of the treated cultures was evaluated and compared to untreated cultures after 21 days by quantification of the weight increase. Kynurenine treatments significantly reduced the proliferation of PEMs in comparison to control cultures by 1.6-fold and 3.6-fold in 200 and 400 µM treatments, respectively (Figure 5B).

The subsequent development of treated cultures was also evaluated and compared to the control cultures. The transfer of the embryogenic masses to the culture medium without kynurenine (recovery conditions) allowed for culture development and the formation of embryos, with very different proportions between the treated and untreated cultures. After 30 days in recovery conditions (Figure 6A,B), the PEMs treated with 200 µM kynurenine showed a reduction, by almost 75%, in the production of mature somatic embryos compared to untreated PEMs, whereas the highest concentration of the inhibitor completely impaired embryo differentiation (Figure 6C). These results suggest a crucial role of auxin in the process.

### 2.3. Expression of Auxin Biosynthesis and Signaling Genes QsTAR2 and QsARF5 during SE and after Treatment with the Inhibitor of Auxin Biosynthesis

To evaluate the dynamics of endogenous auxin before and after SE induction in cork oak, we analyzed the expression of two key genes involved in auxin signaling and auxin biosynthesis. First, we analyzed a *TRYPTOPHAN AMINOTRANSFERASE-RELATED QsTAR2* gene involved in the major tryptophan-dependent pathway of auxin biosynthesis [33]. We also chose the *AUXIN RESPONSE FACTOR QsARF5* gene, suggested to be a key regulator of auxin signaling as it modulates auxin-dependent gene transcription [53]. The analyses were performed in immature zygotic embryos (before SE induction); in embryogenic masses (after SE induction), which include PEMs and the early globular embryos arising from them; and in differentiating somatic embryos, mainly torpedo and cotyledonary embryos. The expression level of the *QsTAR2* gene in immature zygotic embryos was low, but expression progressively increased in embryogenic masses, accompanying embryogenesis induction and embryo development (Figure 7A). With embryo differentiation, *QsTAR2* exhibited the highest expression levels, increasing its expression by 3.6-fold compared to immature zygotic embryos and twice compared to embryogenic masses (Figure 7A). The results showed that *QsARF5* expression was very low before the induction of SE, but it was highly activated after induction, by around 13-fold in embryogenic masses and 11-fold once the differentiation of the somatic embryos took place (Figure 7B). These results demonstrate the activation of auxin biosynthesis and signaling with embryogenesis induction and progression, correlating with immunofluorescence results, which indicated that the auxin content increased with the activation and progression of the SE program.

In order to evaluate the effect of kynurenine on the auxin-related genes in embryogenic cultures, we evaluated how this drug affected the expression of the *QsTAR2* and *QsARF5* genes. Kynurenine drastically reduced the expression of the auxin biosynthesis gene *QsTAR2* in embryogenic masses treated with the inhibitor compared to untreated cultures (Figure 7C). The expression pattern for *QsARF5* was similar to that obtained for *QsTAR2* (Figure 7D). In both cases, kynurenine treatment reduced more than half the expression of the auxin biosynthesis and signaling genes. These results were in consonance with the observations of the effect of this inhibitor in embryogenic cultures, where proliferation was considerably reduced after this treatment.

## 3. Discussion

### 3.1. Endogenous Auxin Accumulation Accompanies SE Induction and Embryo Development

It has been widely reported that auxin acts as a signal required for triggering the transition of somatic cells into an embryogenic program [19,54,55,56]. In many species, treatments with exogenous auxins, such as 2,4-D, are a major requirement for the induction of SE [16,17,39]. However, in some species, removal of the exogenous auxin supply is necessary for SE initiation, as is the case in cork oak. Thus, in the present study, we analyzed to what extent endogenous auxin is activated and plays a role in SE induction and progression. Since the aim of the study was to evaluate the general auxin dynamics during the process, independently of the genotype, SE was induced in immature zygotic embryos collected from several trees of unknown genotype which were randomly chosen from the field. In our work, the IAA immunofluorescence assays showed very low auxin content in the somatic cells of the immature zygotic embryos (before induction), but the accumulation of endogenous auxin in PEMs (after induction) increased considerably. This result agrees with the preliminary observation of Rodriguez-Sanz et al. (2014) [48], in whose study IAA accumulation was detected in embryogenic cells of cork oak PEMs formed from immature zygotic embryos. In cork oak, the application of 2,4-D in the culture medium is needed as a pretreatment for the induction of SE, while the initiation of the new embryogenic program occurs only after auxin’s removal from cultures [11]. It has been proposed that the addition of 2,4-D induces cell reprogramming and, in consequence, an embryogenic response that is associated with the increase in the endogenous levels of IAA [57,58]. Differences in the accumulation of IAA before and after the induction of SE have also been reported in other tree species. In *Quercus alba*, a very low IAA immunofluorescence signal was observed in leaf explants, whereas an intense IAA labeling was detected in PEMs [46]. Moreover, it has been described that in *Quercus alba* [46] and *Solanum betaceum* [59], after induction, endogenous IAA content is higher in embryogenic cells than in non-embryogenic cells.

The present results also reveal that later on in development, during cork oak SE progression, developing somatic embryos (heart-torpedo and early cotyledonary somatic embryos) displayed increased auxin accumulation. It is noteworthy that the meristematic cells of these advanced embryos exhibited high auxin content, which is in relation to the proliferative activity of this kind of cells. Likewise, endogenous auxin levels have been shown to be relatively high in mature somatic embryos of *Picea abies* [44,60] and *Abies alba* [45]. Vondrakova et al. (2018) [60] performed a thorough analysis of endogenous IAA content in the somatic embryos of Norway spruce during SE. These authors observed that the concentration maxima for IAA was detected at the maturation stage of development, suggesting a correlation between this high content and the polarization of somatic embryos. Our results indicate, for the first time in cork oak, that the dynamics of auxin accumulation are similar during SE induction and progression.

### 3.2. Endogenous Auxin Biosynthesis Is Needed for Proliferation of Embryogenic Cells and SE Progression

In this study, a functional analysis was performed by investigating the inhibition of auxin biosynthesis with the small molecule kynurenine, which has not been used in cork oak before. Kynurenine inhibits the enzymatic activity of TAA1/TAR [50]. In cork oak, SE proliferation and later stages of embryo development take place in an auxin-free medium. When kynurenine was added to SE cultures, the proliferation of PEMs was reduced drastically compared to untreated cultures, suggesting that the activation of proliferation could be explained by de novo biosynthesis of auxin. Later in the development, the removal of kynurenine permitted the formation of embryos from PEMs; however, the inhibition of auxin biosynthesis by kynurenine led to a significant reduction in mature somatic embryo production. These results suggest that auxin biosynthesis is necessary not only for cell proliferation of PEMs, but also for embryo differentiation and maturation, developmental processes that were significantly reduced even after kynurenine removal, which may indicate that certain levels of auxin accumulation in PEM cells are essential to initiate the embryogenic program. In a previous report, it was shown that in barley, kynurenine treatment reduced endogenous auxin content in microspore cultures [43]. After induction by a temperature treatment without exogenous auxin application, the microspore, as a totipotent cell, acquired a new cell fate and initiated embryogenesis, a process that began with a proliferation event. In this study on barley, authors pointed out that continuous treatment with the drug reduced embryogenesis initiation and impaired embryo development [43]. In trees, early events in embryogenesis are crucial for the successful development of somatic embryos [61,62]. In *P. abies*, it has been speculated that the presence of auxin at the proliferation and early developmental stages of SE is essential for embryos to develop to the maturation phase [63]. Our results also suggest a crucial role of endogenous auxin, synthetized de novo, at the initial stages of PEMs proliferation and embryogenesis that influences further embryo development and production in cork oak.

### 3.3. Activation of Auxin Biosynthesis and Signaling Genes Are Required for SE Induction, Proliferation, and Progression

Several studies in Arabidopsis have revealed that effects of auxin on the regulation of zygotic embryogenesis are determined by the coordination of complex processes, including auxin biosynthesis, transport and signaling [28,64,65,66]. It is generally accepted that auxin-mediated transition of somatic cells into embryogenic cells is accompanied by the activation of genes that regulate these processes. In the present work, the expression profile of the auxin biosynthesis gene *QsTAR2* showed low transcription in immature zygotic embryos before induction, whereas higher expression levels were detected in PEMs with embryogenesis initiation, and in differentiated embryos. These results correlated with the increase in the cellular accumulation of endogenous IAA, revealed by the immunofluorescence signals. In this study, it was observed that kynurenine drastically reduced PEMs proliferation while also impairing embryo production, as a consequence of the inhibitory effect of this drug on auxin biosynthesis. These results are in consonance with the downregulation of *QsTAR2* in PEMs that were treated with kynurenine. In Arabidopsis SE cultures, it was confirmed that the enzymes TRYPTOPHAN AMINOTRANSFERASE OF ARABIDOPSIS 1 (TAA1) and TRYPTOPHAN AMINOTRANSFERASE-RELATED 1 and 2 (TAR1 and TAR2) control the main auxin biosynthesis pathway [32,33]. Furthermore, 2,4-D treatment activates these core regulators of the auxin biosynthesis pathway in SE-induced explants of Arabidopsis [67]. Thus, our results suggest that the activation of auxin biosynthesis is required for the induction of embryogenic competence and proliferation of embryogenic cells, as well as throughout embryo development. Similar results have been obtained for microspore embryogenesis in *H. vulgare* [43]; the increased expression of the *HvTAR2-like* gene has been reported to participate in the induction of the auxin biosynthesis pathway in microspore embryogenesis, being upregulated at embryogenesis initiation and showing its maximum expression at advanced developmental stages (coleoptilar embryos). In *B. napus*, a low expression of the auxin biosynthesis *BnTAA1* gene was found in vacuolated microspores before embryogenesis induction, and a significant increase was detected in its expression at the early multicellular embryo stage [42]. In *Coffea canephora,* a correlation between an increase in the content of endogenous IAA and in the expression of *CcTAA1* was also reported [68].

Auxin-related responses by cells depend not only on its biosynthesis, but also on the genetic components of the auxin-signaling pathway, where AUXIN RESPONSE FACTORs (ARFs) play a key role in controlling the target gene expression in response to auxin [24]. Our results showed low transcription of *QsARF5* in immature zygotic embryos prior to induction. However, *QsARF5* showed a significant upregulation with embryogenesis initiation and during embryo development and differentiation. The changes in auxin cellular content before and after induction and in embryo differentiation detected by immunofluorescence assays correlated with the expression pattern of this auxin-signaling gene. Furthermore, after kynurenine treatment, *QsARF5* was downregulated considerably in treated PEMs. In Arabidopsis, Wójcikowska and Gaj (2017) [38] reported that the modulation of several ARFs transcripts suggested the extensive participation of auxin signaling during the SE process. Moreover, these authors confirmed that among the 23 *ARF* genes described in this species, *ARF5* played a central role controlling the embryogenic transition induced in somatic cells. In cork oak, Capote et al. (2019) [69] analyzed the expression of the *QsARF5* gene at different developmental stages of SE, from globular to mature somatic embryos. They detected the activation of this gene in the early stages of cork oak SE, specifically at the globular SE stage, suggesting the possible involvement of *QsARF5* in the control of cell identity and embryo differentiation. These findings clearly indicate that auxin biosynthesis and signaling are required for the proper development of embryogenic cultures.

## 4. Materials and Methods

### 4.1. Somatic Embryogenesis Cultures

SE was induced from immature zygotic cork oak embryos [10], following the updated protocol described by Testillano et al. (2018) [11]. Immature pollinated acorns at the responsive stage of early cotyledonary embryos (around September) were collected from 4 selected trees of random origin located in El Pardo Forest, Madrid, Spain. A pool of randomly chosen immature acorns was used to induce the embryogenic response, not taking into account the trees’ genetic backgrounds. Immature acorns were cultivated in induction media supplemented with the plant growth regulator 2,4-D (Sigma-Aldrich, Saint Louis, MO, USA), at 25 °C, with 16/8 h light/darkness. After 30 days, they were transferred to proliferation media [11], a regulator-free medium where proembryogenic masses (PEMs) and somatic embryos developed. Cultures were transferred monthly to a fresh medium of the same type, and SE cultures continued their development and multiplied, producing new PEMs and somatic embryos.

### 4.2. Sample Fixation and Processing for Microscopy

Samples from immature zygotic embryos and different developmental stages of the SE process (PEMs, heart and torpedo somatic embryos) were collected and fixed with 4% paraformaldehyde in phosphate-buffered saline (PBS) (pH 7.3) overnight at 4°C. Then, fixed samples were washed in PBS, dehydrated in acetone series (30%, 50%, 70%, 90%, and 100%), embedded in Technovit 8100 acrylic resin (Kulzer GmbH, Wehrheim, Germany), and polymerized at 4 °C. Semithin sections of 2 µm thickness were either stained with Toluidine blue and observed under a bright-field microscope for cellular structure analysis, or placed on APTES-coated slides and kept at 4 °C until use for immunofluorescence assays.

### 4.3. Immunofluorescence and Laser Confocal Microscopy Analysis

Semithin sections of immature zygotic embryos, PEMs, and heart and torpedo embryos were blocked by 10% (*w*/*v*) fetal calf serum (FCS) in PBS for 10 min, washed in 1% PBS, and incubated for 1 h with anti-IAA mouse monoclonal antibody (Sigma, cat. no. A0855) which had been diluted 1:100 in 1% (*w*/*v*) bovine serum albumin (BSA) in PBS. After washing in 1% PBS, the signal was revealed with Alexa Fluor 488-labeled anti-mouse IgG antibody (Molecular Probes, Eugene, OR, USA) which had been diluted 1:25 in 1% (*w*/*v*) BSA in PBS for 45 min in darkness. Afterwards, sections were washed in 1% PBS, counterstained with 1 mg/mL 4, 6-diamidino-2-phenylindole dihydrochloride (DAPI) for 10 min, and washed again in 1% PBS. Finally, sections were mounted in Mowiol and analyzed using a confocal laser microscope (Leica TCS-SP5-AOBS, Vienna, Austria). Maximum projection images were obtained using confocal microscopy software (Leica software LCS version 2.5). The confocal microscopy analysis was performed using the same laser excitation and sample emission capture settings for image acquisition for all immunofluorescence preparations, allowing an accurate comparison to be made among signal intensities. Controls were created by omitting the primary antibody in the immunofluorescence assay.

### 4.4. Treatment with the Inhibitor of Auxin Biosynthesis Kynurenine

The effect of kynurenine (Sigma-Aldrich, Saint Louis, MO, USA) was evaluated on PEMs that originated from randomly selected immature zygotic embryos. The inhibitor was added to proliferation media from a freshly prepared stock solution of 10 mM in dimethyl sulfoxide (DMSO). Stock solution was added to the media after filtering with a sterile Ministart^®^ filter (Sartorius Stedim Biotech, Goettingen, Germany). Different concentrations of kynurenine, i.e., 200 and 400 µM, were assayed. PEMs were cultured under these conditions for 21 days, and 4 replicates per concentration were assayed. Parallel plates without the drug were used as controls. After 21 days of treatment, PEMs were transferred to a recovery medium that consisted of the proliferation medium without the inhibitor. The effect of kynurenine on SE was assessed by quantifying the proliferation of the PEMs by relative fresh weight (FW) (FW after 21 days of treatment/FW at the initiation of the experiment), and by quantifying the percentage of somatic embryos produced after 30 days in the recovery medium. Differences between the treated and control cultures were tested by ANOVA (analysis of variance) and Tukey’s tests at *p* ≤ 0.05.

### 4.5. Gene Expression Analysis by RT-qPCR

Expression analyses of the *TRYPTOPHAN AMINO TRANSFERASE-RELATED PROTEIN 2* (*QsTAR2;* (accession number: XM_024022178.1)) and *AUXIN RESPONSE FACTOR 5* genes (*QsARF5*; (accession number: XM_024044312.1)) were performed. Sequences were selected from the NCBI database, www.ncbi.nlm.nih.gov/genbank (accessed on 1 May 2022). The analyses were carried out on immature zygotic embryos (before induction) and at different stages of development of SE cultures (after induction) (Figure 1): embryogenic masses and differentiating embryos (isolated heart and torpedo embryos). In addition, the expression levels of *QsTAR2* and *QsARF5* were analyzed for untreated and 200 µM kynurenine-treated embryogenic masses. Total RNA from the samples was purified with the NucleoSpin^®^ RNA Plant (Macherey-Nagel, Düsen, Germany) according to the manufacturer’s instructions. RAP buffer with 1% β-mercaptoethanol was used. Contaminated DNA was removed from the total RNA samples using the Turbo DNA-freeTM Kit (Ambion, Life Technologies, Carlsbad, CA, USA) according to the supplier’s protocol. A 300 ng aliquot of total RNA was used for the reverse transcription reaction using the SuperscriptTM II Reverse Transcriptase (Invitrogen Life Technologies, Carlsbad, CA, USA) according to the manufacturer’s instructions. Gene-specific primers were designed using Primer3 software [70] with default parameters and amendments according to the following criteria: melting temperature around 70 °C and product size between 80 and 170 bp. The oligonucleotides used were, for *QsTAR2* FW, 5′-TACAGTCTCAAAGAGCACGGG-3′; for *QsTAR2* RW, 5′-CAACTTCCACCTCTCTGCCA-3′; for *QsARF5* FW, 5′-GAAGCCCCACCTCCTAGATTC-3′; and for *QsARF5* RW, 5′-TTCCCTGTCCCCCATTACTC-3′.

RT-qPCR was performed using the FastStart Essential DNA Green Master (Roche Diagnostics, Indianapolis, IN, USA) on the LightCycler^®^96 (Roche Diagnostics International Ltd.). Thermocycle settings were carried out as follows: initial denaturation for 30 s at 95 °C, followed by 40 cycles, each consisting of 5 s, at 95 °C and 30 s at 58 °C. After each run, a dissociation curve was acquired to check for amplification specificity by heating the samples from 58 to 95 °C. As internal reference gene, *ACTIN* (*QsACTIN*; accession number: EU697020.1), was used. A minimum of three biological and three technical replicates were analyzed. Samples of immature zygotic embryos were extracted from a random pool of at least 4 immature acorns. Samples of each developmental stage were randomly extracted from at least 8 different SE cultures. Samples of kynurenine treatments and controls were randomly extracted from 3 different SE cultures. Data were analyzed using the LightCycler^®^96 software (v.1.1.0.1320) (Roche Diagnostics International Ltd.), and the Livak calculation method [71]. Transcript levels were normalized using *QsACTIN* values. Data are expressed as mean values of relative expression (fold-change values) to PEMs during SE progression and to control cultures in kynurenine study. Differences were tested by one-way ANOVA analysis of variance followed by Tukey’s multiple analysis test at *p* ≤ 0.05.

## 5. Conclusions

In summary, the presented results illustrate the cellular auxin accumulation dynamics during SE in cork oak, and reveal that endogenous auxin levels and the expression of auxin biosynthesis and signaling genes increase after SE induction. During further somatic embryo development, auxin accumulation in the cells gradually increases, together with the expression of *QsTAR2* biosynthesis gene, while *QsARF5* signaling gene maintains very high expression in PMEs and developing embryos. Furthermore, kynurenine-mediated inhibition of auxin biosynthesis significantly affects proliferation and differentiation events, indicating the requirement of de novo auxin biosynthesis for correct in vitro embryo development. Taken together, these findings suggest that in cork oak, auxin leads the transition of somatic cells towards an embryogenic program by the coordination of auxin biosynthesis and signaling processes. This information provides new insights into the regulating mechanisms of SE in forest species, where data are still scarce, opening the door for novel strategies through selective targets for improving the efficiency of the process in tree breeding programs.

## Figures and Tables

**Figure 1 plants-12-01542-f001:**
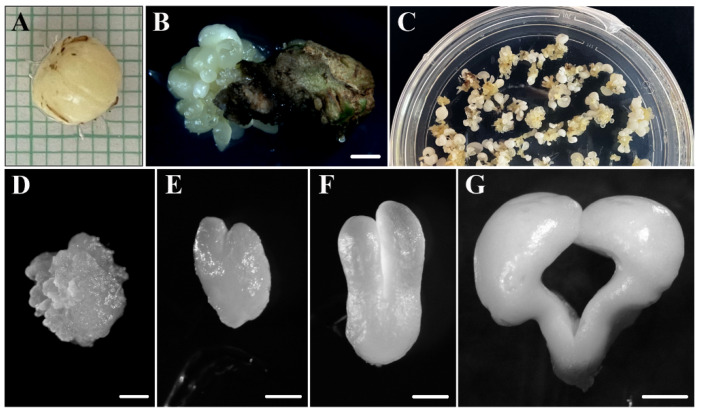
Main stages of somatic embryogenesis of *Quercus suber*. (**A**) Immature zygotic embryos. (**B**) Induction of somatic embryos from an immature zygotic embryo. (**C**) Panoramic view of a culture plate showing different structures corresponding to various developmental stages. (**D**) Proembryogenic mass. (**E**) Heart-shaped embryo. (**F**) Torpedo embryo. (**G**) Mature cotyledonary embryo. Bars in: (**B**) 2 mm, (**C**) 1 cm, (**D**–**F**) 0.5 mm, and (**G**) 1 mm.

**Figure 2 plants-12-01542-f002:**
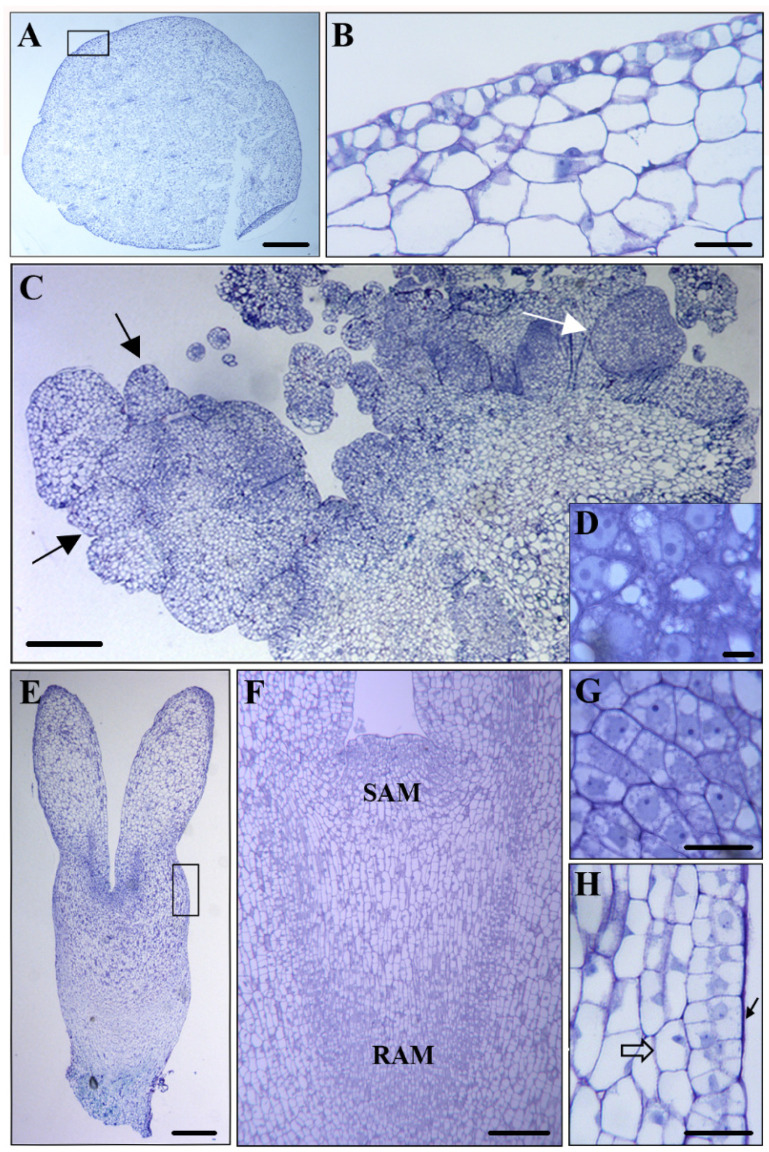
Cellular organization of main stages of somatic embryogenesis. Micrographs of semithin sections stained by Toluidine blue. (**A**,**C**,**E**) Panoramic view of immature zygotic embryo, (**A**) cluster of proembryogenic masses, and (**C**) torpedo embryo (**E**). (**B**,**D**,**F**,**G**,**H**) Detail of representative regions at higher magnification. (**B**) Epidermis of immature zygotic embryos, as indicated by the square in (**A**). (**D**) Characteristic cells of embryogenic masses, with large central nuclei and prominent nucleoli. (**F**) Shoot apical meristem (SAM) and root apical meristem (RAM) of torpedo embryo. (**G**) Meristematic cells of SAM. (**H**) Differentiating epidermis of torpedo embryo, as indicated by the square in (**E**). Black arrows represent globular embryos in (**C**) and polygonal cells of epidermis in (**H**). White arrow represents heart-shaped embryo in (**C**). Open arrow represents differentiated cells in (**H**). Bars in: (**A**,**E**) 0.5 mm, (**B**,**G**,**H**) 20 µm, (**C**,**F**) 250 µm, and (**D**) 5 µm.

**Figure 3 plants-12-01542-f003:**
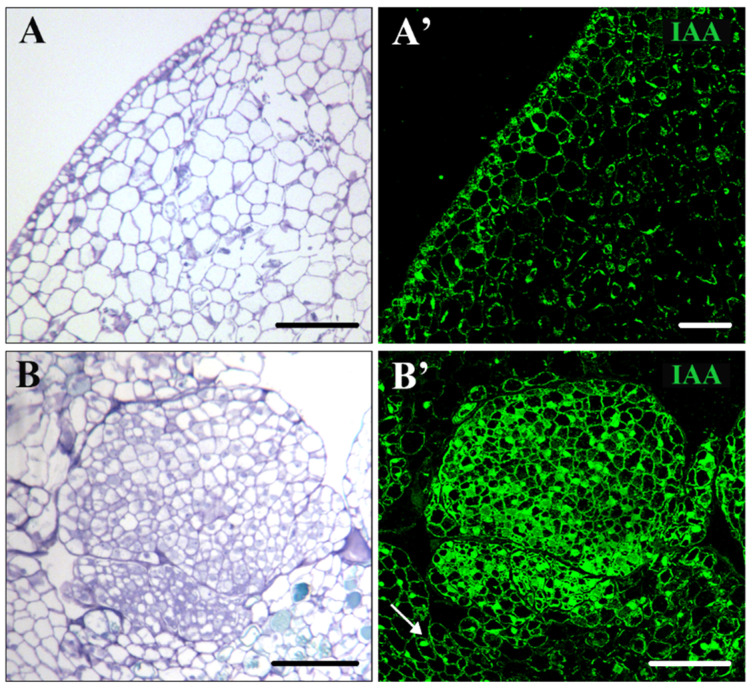
Immunolocalization of IAA before and after SE induction in immature zygotic embryos. (**A**,**B**) Micrographs of semithin sections stained by Toluidine blue of immature zygotic embryo (**A**) and proembryogenic masses (B). (**A’**,**B’**) Confocal microscopy images of IAA immunofluorescence (green signal) of immature zygotic embryo (**A’**) and proembryogenic mass (**B’**). Thin arrow points to vacuolated cells that surrounds the embryogenic cell cluster. Bars represent: (**A**,**A’**) 50 µm and (**B**,**B’**) 75 µm.

**Figure 4 plants-12-01542-f004:**
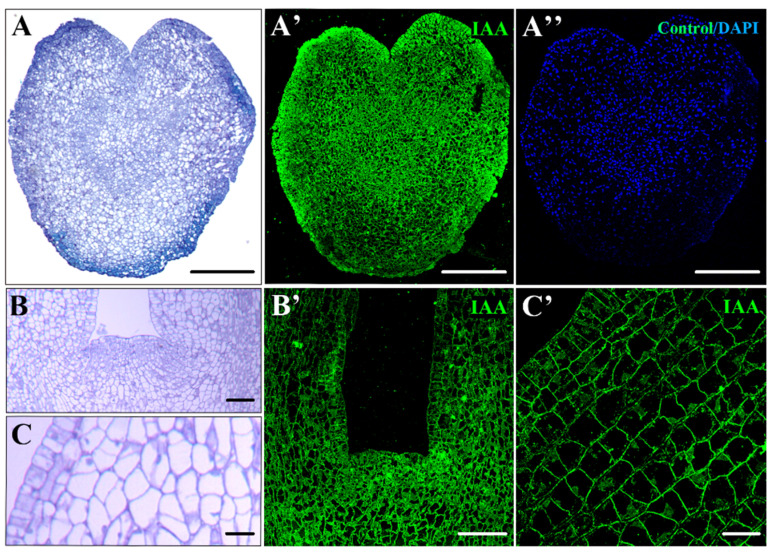
Immunolocalization of IAA at advanced stages of development of somatic embryos. (**A**,**B**,**C**) Micrographs of semithin sections stained by Toluidine blue: the heart (**A**); shoot apical meristem (**B**); and cotyledon of the torpedo embryo (**C**). (**A’**,**B’**,**C’**) Confocal microscopy images of IAA immunofluorescence (green signal) of the heart (**A’**); shoot apical meristem (**B’**); and cotyledon of the torpedo embryo (**C’**). (**A”**) Negative control omitting the primary antibody. Bars represent: (**A**,**A’**,**A”**) 250 µm, (**B**) 125 µm, (**B’**) 75 µm, (**C**,**C’**) 25 µm.

**Figure 5 plants-12-01542-f005:**
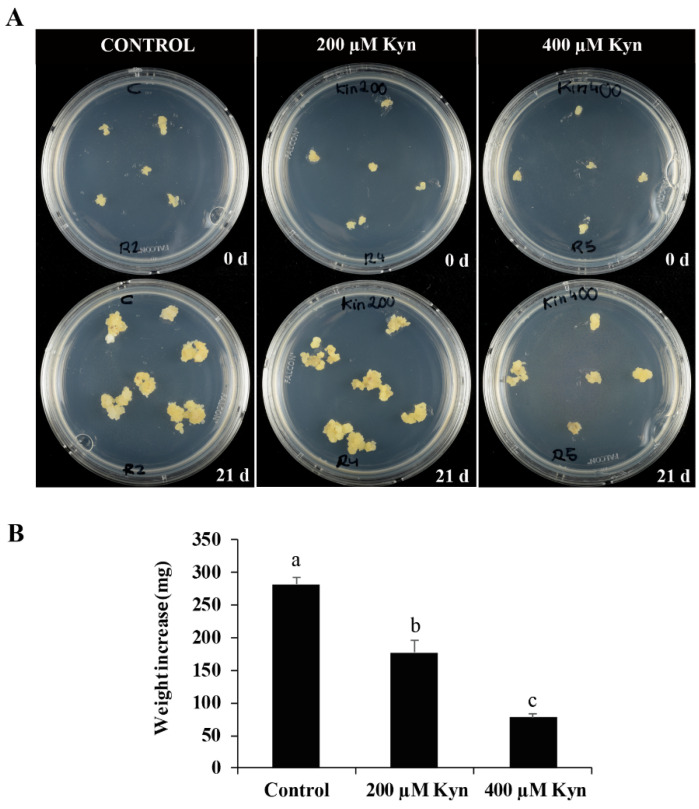
Effect of the inhibitor of auxin biosynthesis kynurenine on SE. (**A**) SE cultures at the beginning of the treatment, at 0 days, and after 21 days of treatment. Left pictures: untreated cultures; central pictures: 200 µM kynurenine treatment; right pictures: 400 µM kynurenine treatment. (**B**) Quantification of the proliferation of proembryogenic masses in control cultures and cultures treated with kynurenine. Columns represent mean values of relative fresh weight after 21 d of treatment; bars represent the standard error of the mean. Different letters in the columns indicate significant differences according to ANOVA and Tukey’s test at *p* ≤ 0.05.

**Figure 6 plants-12-01542-f006:**
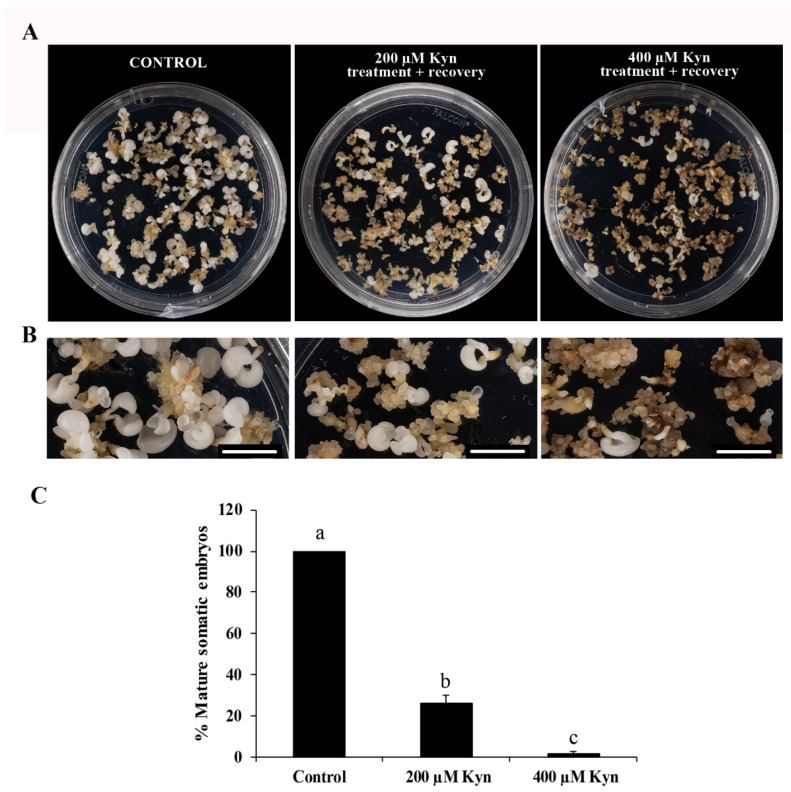
Effect of kynurenine on the recovery of SE cultures. (**A**) Cultures after 30 days in recovery conditions. Left picture: control cultures; central picture: 200 µM kynurenine-treated cultures in recovery; right picture: 400 µM kynurenine-treated cultures in recovery. (**B**) Detail at higher magnification of cultures after 30 days in recovery conditions. Left picture: control cultures; central picture: 200 µM kynurenine-treated cultures in recovery; right picture: 400 µM kynurenine-treated cultures in recovery. (**C**) Quantification of embryos produced after 30 days in recovery conditions in control cultures and cultures treated with kynurenine. Columns represent mean values of the percentage of mature somatic embryos; bars represent the standard error of the mean. Different letters in the columns indicate significant differences according to ANOVA and Tukey’s test at *p* ≤ 0.05. Bars represent 1 cm.

**Figure 7 plants-12-01542-f007:**
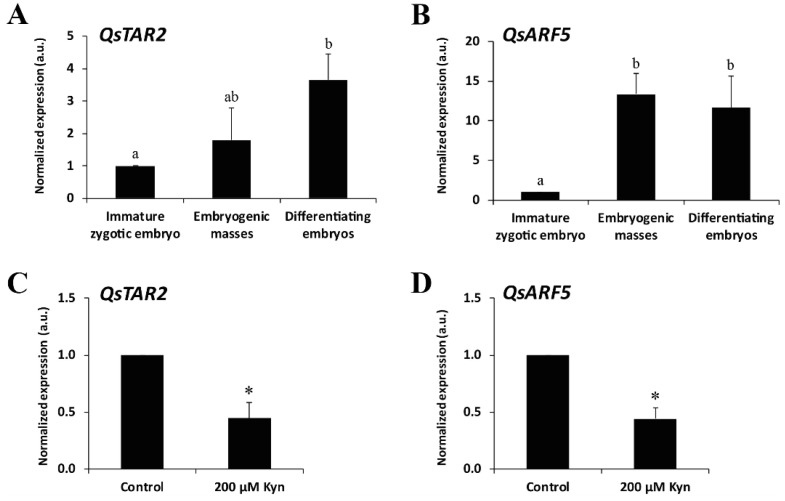
Gene expression patterns of auxin biosynthesis *QsTAR2* and auxin signaling *QsARF5* genes by RT-qPCR, during somatic embryogenesis (**A**,**B**) and in control and kynurenine-treated proembryogenic masses after 21 days (**C**,**D**). Each column represents the mean of at least three biological and three technical replicates. Transcript levels were normalized to *QsACTIN*. Data were expressed as mean values of relative expression to immature zygotic embryos in (**A**,**B**) and to control cultures in (**C**,**D**); bars represent the standard error of the mean. Different letters and asterisks in columns indicate significant differences according to ANOVA and Tukey’s test at *p* ≤ 0.05.

## Data Availability

The data presented in this study are available on request from the corresponding author.

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
