# Peer review of "Dynamics of Endogenous Auxin and Its Role in Somatic Embryogenesis Induction and Progression in Cork Oak"

_plants, 2023, doi:10.3390/plants12071542_

Round 1
Reviewer 1 Report
The ms is focused with clearly presented results supported by appropriate discussion. It complements the earlier studies on the topic and adds to and strengthens the current views on the role of auxin in somatic embryogenesis.
The ms is well written but the M & M requires some more information related to the used material. Please add information on how many different genotypes were used in the experiments, and how potential genotypic variation was taken into account. This is important knowing how much the reponse of various genotypes can vary.
* Obviously, at least zygotic embryos were of different genotypes - and it would be good to tell if they shared the same genetic background (donor tree or controlled crossing) or if they were of more random origin ?
* When zygotic embryos were sampled for gene expression analysis, how many embryos were used per a sample ? I.e. if the results represent a single genotypes or are derived from a combained sample ?
* Likewise, describe if there were several genotypes in the SE cultures used to compare e.g. different developmental phases or kyurenine effect etc., and if care was taken to make comparisons within a genotype ? E.g. not to compare PEMs of one genotype with more developed SE of another genotype... If the genotype effect was not taken into account, give reasoning for that.
* Further, if several genotypes were included, please tell in the results if there were any differences seen among them. In this case, also add discussion on the magnitude and potential significance of genotypic variation
Other minor remarks include adding scientific name of cork oak in the abstract (r18) and writing out NPA and PCIB in the introduction (r91)
Author Response
ANSWER TO REVIEWER 1:
Thank you for your positive comments. We have revised the manuscript taking into account all your concerns, which have been addressed in the revised version. They have been very useful not only to make an improved version of the manuscript but also to be considered for future. studies. Specific responses to your comments and the modifications included in the revised manuscript are listed below.
REVIEWER
The ms is well written but the M & M requires some more information related to the used material. Please add information on how many different genotypes were used in the experiments, and how potential genotypic variation was taken into account. This is important knowing how much the reponse of various genotypes can vary.
Obviously, at least zygotic embryos were of different genotypes - and it would be good to tell if they shared the same genetic background (donor tree or controlled crossing) or if they were of more random origin?
ANSWER
The aim of the study was to evaluate the general auxin dynamics during the process, independently of the genotype, therefore, SE was induced in immature zygotic embryos collected from several (4) donor trees of unknown genotype, randomly chosen in the field, not coming from controlled crossing.
We have included this information in Material & Methods, in the paragraph “4.1. Somatic embryogenesis cultures”, as well as in the 1st paragraph of Discussion.
REVIEWER
When zygotic embryos were sampled for gene expression analysis, how many embryos were used per a sample? I.e. if the results represent a single genotypes or are derived from a combained sample?
ANSWER
For gene expression analyses, RNA samples of immature zygotic embryos were extracted from a random pool of at least 4 immature acorns collected from 4 trees randomly chosen in the field. Therefore, the results are not derived from any specific genotype. We have included this information in the paragraph “4.5. Gene expression analysis by RT-qPCR”, in Material & Methods.
REVIEWER
Likewise, describe if there were several genotypes in the SE cultures used to compare e.g. different developmental phases or kyurenine effect etc., and if care was taken to make comparisons within a genotype? E.g. not to compare PEMs of one genotype with more developed SE of another genotype... If the genotype effect was not taken into account, give reasoning for that.
ANSWER
As described above, the effect of genotype was not considered in this study. The reason is that the aim of the study was to evaluate the general auxin dynamics during the process, independently of the genotype, therefore, the use of pools of random and combined samples was convenient.
REVIEWER
Further, if several genotypes were included, please tell in the results if there were any differences seen among them. In this case, also add discussion on the magnitude and potential significance of genotypic variation.
ANSWER
As described above, the effect of genotype was not considered in this study.
REVIEWER
Other minor remarks include adding scientific name of cork oak in the abstract (r18) and writing out NPA and PCIB in the introduction (r91).
ANSWER
Quercus suber L. has been included in line 18 as well as the complete names for NPA and PCIB in line 91.
Moreover, we have performed a final check of the English language to avoid any minor spelling/grammar mistakes.
We hope that now, the revised version of the manuscript will be acceptable for publication.

Reviewer 2 Report
Dear Authors,
Manuscript: Endogenous auxin dynamic and its role in somatic embryogenesis induction and progression in cork oak.
Review: Very interesting and well-written ms describing inside is SE in oak with nice histological data supplemented with immunofluorescent localization of auxin during SE development including also a very interesting part of gene expression and result with auxin inhibitors.
This publication can be inspiring for other researchers working with the somatic embryogenesis of different tree species.
1 Introduction
Well written.
2 Results
Well-written and documented results in all sections, just pictures ABC, from Fig. 1 can be better quality.
Otherwise, very nice Fig 2 documenting histology and Fig. 3 and Fig.4 immunofluorescent study in this interesting ms.
Unfortunately, Fig. 5 and Fig. 6 are not of good quality, the structure of the embryogenic calli is hardly visible, and can be improved. They are supposed to show a very interesting part of the study with kynurenine and it is pity that these results are not well documented here.
Discussion
Well-written and nice reading.
Conclusion
Well written and clearly formulated. But suppose to be a new paragraph and not part of the Discussion.
M&M
Well written.
20.3.2023
Author Response
ANSWER TO REVIEWER 2
Thank you for your positive comments. We have revised the manuscript taking into account all your concerns, which have been addressed in the revised version. Specific responses to your comments and the modifications included in the revised manuscript are listed below.
Figures 1, 5 and 6 have been replaced by images with better quality and resolution, in TIF format.
- Furthermore, Figure 6 has been reorganized to add new images that correspond to higher magnification details of the culture plates, in order to better illustrate the results on the development of the embryogenic cultures after kynurenine treatment.
- Conclusions have been included in a new paragraph, after Material and Methods.
Moreover, we have performed a final check of the English language to avoid any minor spelling/grammar mistakes.
We hope that now, the revised version of the manuscript will be acceptable for publication.
